# A novel machine learning algorithm selects proteome signature to specifically identify cancer exosomes

Bingrui Li[1], Fernanda G Kugeratski[1], Raghu Kalluri[1,2,3]*

[1]Department of Cancer Biology, University of Texas MD Anderson Cancer Center, Houston, United States; [2]Department of Bioengineering, Rice University, Houston, United States; [3]Department of Molecular and Cellular Biology, Baylor College of Medicine, Houston, United States

**Abstract** Non-invasive early cancer diagnosis remains challenging due to the low sensitivity and specificity of current diagnostic approaches. Exosomes are membrane-bound nanovesicles secreted by all cells that contain DNA, RNA, and proteins that are representative of the parent cells. This property, along with the abundance of exosomes in biological fluids makes them compelling candidates as biomarkers. However, a rapid and flexible exosome-based diagnostic method to distinguish human cancers across cancer types in diverse biological fluids is yet to be defined. Here, we describe a novel machine learning-based computational method to distinguish cancers using a panel of proteins associated with exosomes. Employing datasets of exosome proteins from human cell lines, tissue, plasma, serum, and urine samples from a variety of cancers, we identify Clathrin Heavy Chain (CLTC), Ezrin, (EZR), Talin-1 (TLN1), Adenylyl cyclase-associated protein 1 (CAP1), and Moesin (MSN) as highly abundant universal biomarkers for exosomes and define three panels of pan-cancer exosome proteins that distinguish cancer exosomes from other exosomes and aid in classifying cancer subtypes employing random forest models. All the models using proteins from plasma, serum, or urine-derived exosomes yield AUROC scores higher than 0.91 and demonstrate superior performance compared to Support Vector Machine, K Nearest Neighbor Classifier and Gaussian Naive Bayes. This study provides a reliable protein biomarker signature associated with cancer exosomes with scalable machine learning capability for a sensitive and specific non-invasive method of cancer diagnosis.

## eLife assessment

This **important** study introduces a novel AI method for the analysis of published data, with practical implications for early cancer diagnosis. The results are supported by **compelling** evidence.

## Introduction

Tissue biopsies have traditionally been a definitive way to diagnose and stage cancer; however, a biopsy may not be easily accessible for many tumors such as those in the pancreas, lung, and brain (*Distler et al., 2014*; *Morris et al., 2010*; *Suram et al., 2012*; *Mullerat et al., 2003*; *Hammes et al., 2008*). In addition, the small amount of biopsied tissue does not represent the entire heterogeneous pathological profile of the tumor (*Oshi et al., 2021*). In recent years, liquid biopsy has emerged as a plausible diagnostic and monitoring approach with the capability to detect tumor biomarkers in more accessible biological fluids such as plasma, serum and urine (*Hu et al., 2022*). Detectable tumor

*For correspondence:
rkalluri@mdanderson.org

biomarkers can include circulating tumor DNA (ctDNA), circulating tumor cells (CTCs), and exosomes (***Kalluri and LeBleu, 2020***).

Exosomes are extracellular vesicles of endosomal origin that are between ~40 and 180 nm in diameter and have been shown to mediate intercellular communication in health and disease (***Kalluri and LeBleu, 2020***; ***Johnstone et al., 1987***; ***Maas et al., 2017***). They can contain a variety of biomolecules including DNA, RNA, proteins, lipids, metabolites and other materials representative of the parent cell (***Kalluri and LeBleu, 2020***). Exosomes are present at high concentrations in biological fluids, which is a potential advantage as a biomarker (***Colombo et al., 2014***; ***Valencia and Montuenga, 2021***). Exosomal mRNAs and miRNAs have been investigated intensively as diagnostic biomarkers, and mounting evidence suggests that exosomal proteins circulating in biological fluids could be used for cancer diagnosis and monitoring cancer progression (***Hu et al., 2022***; ***Chen et al., 2017***). Challenges that remain include standardization of methods for consistent exosome isolation from various tissues, identification of biomarkers that distinguish cancer and normal exosomes across different cancer types, and the identification of biomarkers that are unique to specific biological fluids (e.g. plasma, serum, urine, etc.).

In addition to accurately identifying exosomal protein biomarkers, it is also challenging to successfully utilize them to diagnose cancers due to intratumor and interpatient heterogeneity (***Muluneh et al., 2014***). Conventional diagnostic approaches predominantly rely on a single biomarker, which is often not specific or sensitive (***Ballehaninna and Chamberlain, 2011***; ***Singh et al., 2011***). However, recent advances in machine learning algorithms connected to artificial intelligence provide an opportunity to construct a classifier that identifies a panel of exosomal protein biomarkers that would possess a more comprehensive ability to reflect the complex disease status of different patients and distinguish cancer samples from normal samples with significantly improved sensitivity and specificity.

## Results

### Unbiased proteomics analysis of exosomes identifies 18 abundant plasma membrane protein markers for various human cell lines

To identify universal exosomal protein biomarkers for differentiating cancer from non-cancer exosomes, we analyzed protein abundance data from 228 cancer and 57 control cell-line-derived exosomes, representing various cancer types (***Figure 1A***; ***Supplementary file 1***). Because studies employ distinct isolation and mass spectrometry quantification techniques, the number of identified

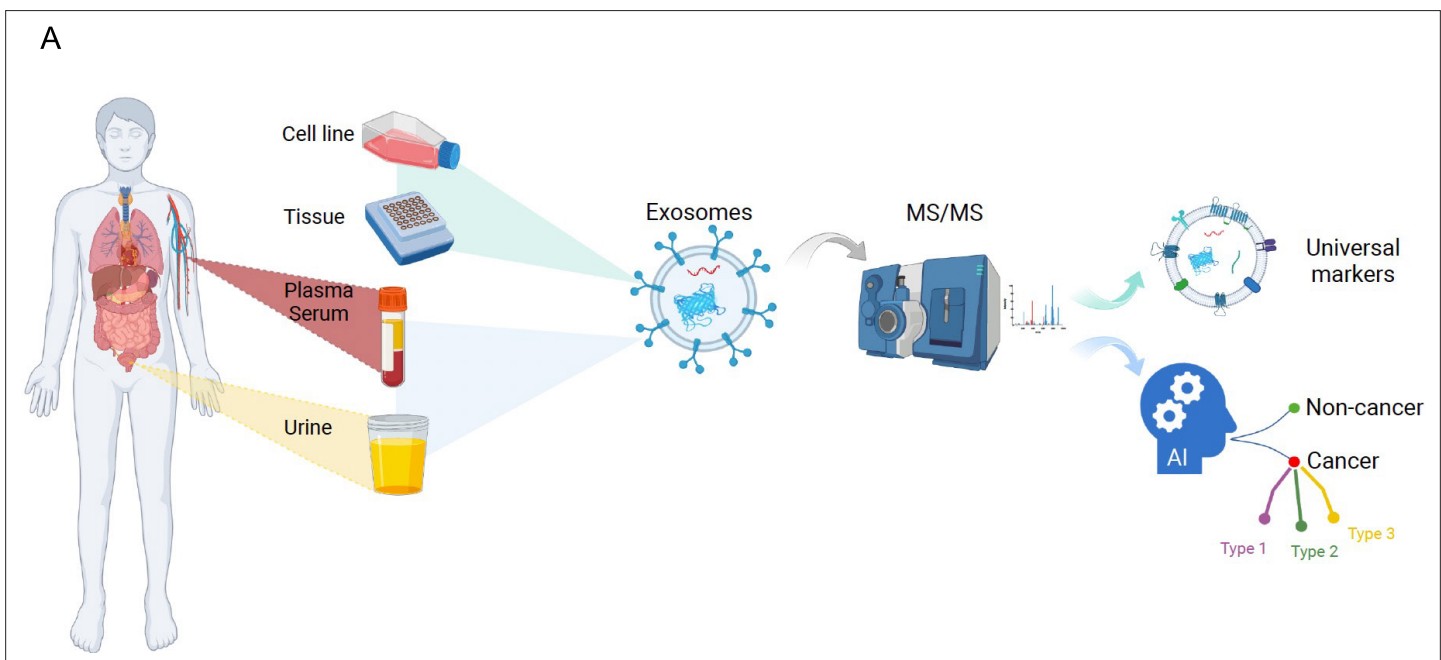

**Figure 1.** Overview of the study.

proteins is different among the studies. To overcome the bias caused by such technical factors, we examined the proteins common to all studies and identified 1124 overlapping proteins (*Figure 2A*). To determine the heterogeneity among cancer and control cell-line-derived exosomes, we performed principal component analysis (PCA) using these 1124 proteins in 285 cancer and control cell line-derived exosomes (*Figure 2B*). The PCA indicated that the exosomes derived from cancer and control cell lines are heterogeneous and show significant variation in protein expression across cell lines.

We next investigated the frequency of the proteins detected in the exosomes from all cell lines, only the cancer cell lines and just the control cell lines. Commonly used exosome biomarkers (e.g. CD9, tetraspanins) were examined first, and from the 12 traditional exosome markers (*Théry et al., 2006*; *Kowal et al., 2016*; *Jeppesen et al., 2019*), only eight proteins were detected among the 1124 overlapping proteins from all cell lines. Six of eight proteins were detected in at least 90% of all samples, with CD9 and HSPA4 detected with the least frequency (*Figure 2C*). In addition, the frequency of FLOT1, FLOT2, and TSG101 proteins was higher in cancer cell-line-derived exosomes when compared to control cell-line-derived exosomes (*Figure 2C*).

To identify biomarkers detectable at a high frequency in all cancer and control cell-line-derived exosomes, we searched for the proteins that were detected in ≥90% of all samples (*Supplementary file 2*). We annotated these proteins using Ingenuity Pathway Analysis (IPA) and found that 78.0% of these proteins localize to the cytoplasm and 13.5% of proteins are associated with the plasma membrane (*Figure 2D*). Gene Ontology (GO) analysis revealed the enrichment of proteins from pathways related to vesicle-mediated transport, secretory vesicles, exocytosis, endocytosis, and other exosome-related pathways (*Figure 2E*). To further explore the utility of the proposed biomarkers, we examined the proteins located on the plasma membrane that met the threshold of detection in ≥90% of all samples (*Figure 2F*). Clathrin Heavy Chain (CLTC) was ranked as the top plasma membrane protein detected in 99.6% of all samples and 100% of control samples (*Figure 2F*). In addition, the scaffolding protein, Syntenin-1(SDCBP) was detected at a high frequency of 97.9% of all samples, corroborating previous findings (*Kugeratski et al., 2021*). Next, we sought to identify unique markers that can identify cancer cell-derived exosomes (cancer exosomes) by filtering out the proteins present in ≤10% of 57 control cell-line-derived exosomes. Interestingly, Ataxin 2 Like (ATXN2L), which has been reported to promote cancer cell invasiveness and resistance to chemotherapy, was uniquely detected in the cancer-cell-derived exosomes (*Figure 3A*; *Lin et al., 2019*). In total, we identified a set of 18 exosome protein markers that are present at a high abundance in all exosomes examined (*Figure 2F*).

## Comparison of exosomal proteins derived from cell lines and tissues identified five universal plasma membrane protein markers

We next sought to investigate common biomarkers for the tissue-derived exosomes across cancer types. We calculated the detection frequency of commonly used exosome markers in the 157 samples (101 cancers; 56 controls). Two established exosome markers, CD63 and TSG101, were only detected in 33.1% and 45.9% of all samples, respectively (*Figure 3B*). To identify high-frequency biomarkers for exosomes derived from both cell lines and tissues, we examined the overlapping proteins that met a threshold of ≥90% in all samples for exosomes derived from cell lines and tissues and found 31 common proteins (*Figure 3C*). Among the 31 proteins, there were five proteins that were detected in over 90% of all cell line and tissue-derived exosomes (*Figure 3D*). These include Clathrin Heavy Chain (CLTC), Ezrin, (EZR), Talin-1 (TLN1), Adenylyl cyclase-associated protein 1 (CAP1), and Moesin (MSN).

## An exosome proteome signature of 18 proteins can differentiate cancer exosomes from non-cancer exosomes across multiple cancer types

Plasma or serum is the most readily accessible source for non-invasive biopsies. We next sought to identify if exosome proteins in plasma and serum could differentiate cancer exosomes from non-cancer exosomes across multiple cancer types. We pooled exosome proteomics data derived from plasma or serum for 205 cancer and 51 control samples from five different studies (*Hoshino et al., 2020*; *Vykoukal et al., 2017*; *Li et al., 2021*; *Lin et al., 2022*; *Hallal et al., 2020*), which included breast cancer, colorectal cancer, glioblastoma, lung carcinoma, liver cancer, neuroblastoma, and pancreatic cancer (*Supplementary file 1*). To account for differences in the methodologies of

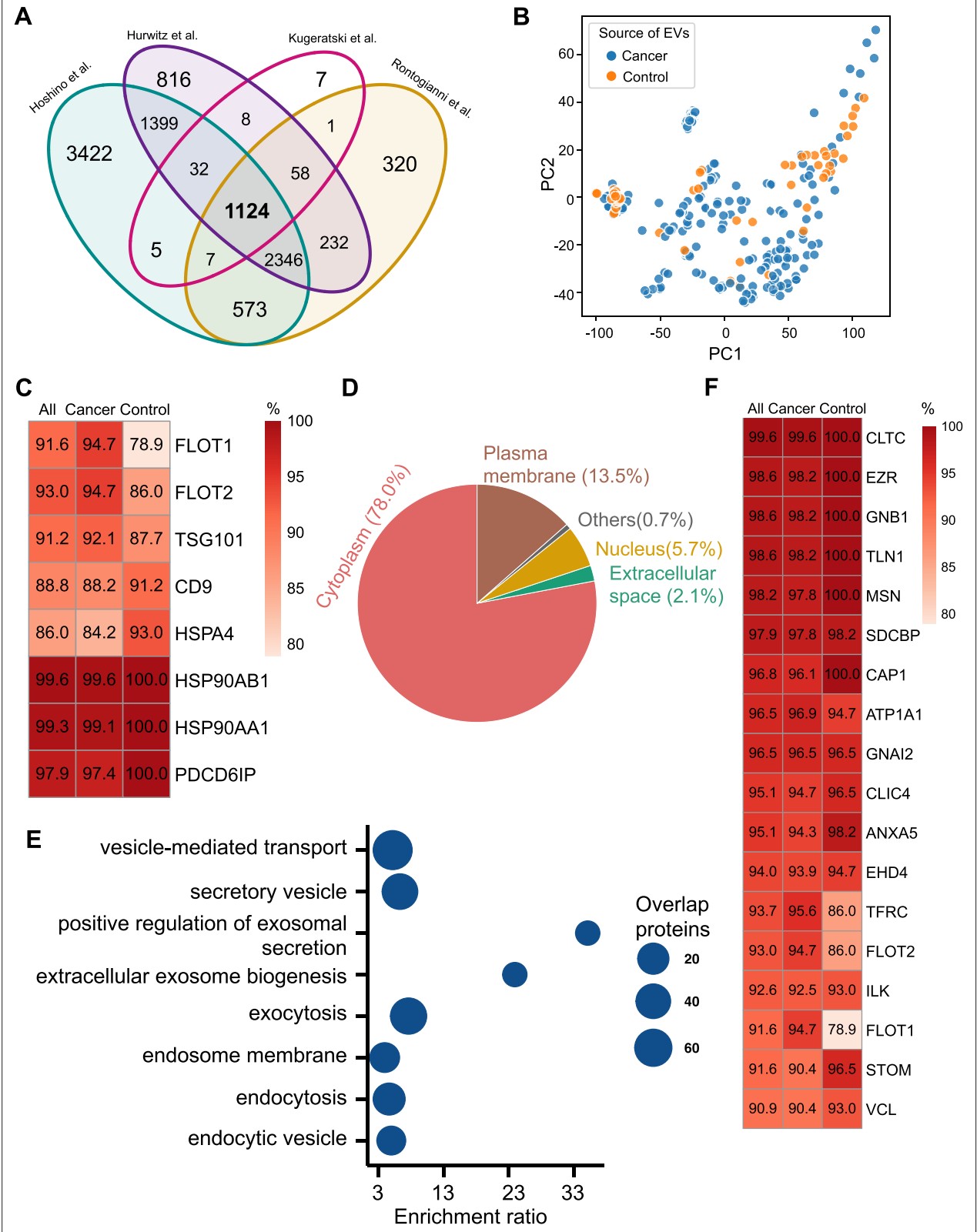

**Figure 2.** Proteomic characterization of exosomes derived from 285 cell lines from four studies. (**A**) Overlapped proteins from four different studies of cell-line-derived exosomes. (**B**) PCA plot of cancer and control cell line-derived exosomes. (**C**) Positivity for eight commonly used exosomal protein biomarkers in various cell lines. The percentage of samples expressing each protein is shown in the boxes. Darker red indicates a higher percentage. (**D**)

*Figure 2 continued on next page*

*Figure 2 continued*

Annotation of the proteins detected in more than 90% of all samples. (**E**) GO and KEGG pathway enrichment analysis of the proteins detected in more than 90% of all samples. (**F**) Plasma membrane proteins detected in more than 90% of all samples.

The online version of this article includes the following source data for figure 2:

**Source data 1.** Related to *Figure 2A*.

**Source data 2.** Related to *Figure 2B*.

**Source data 3.** Related to *Figure 2C and F*.

**Source data 4.** Related to *Figure 2D*.

**Source data 5.** Related to *Figure 2E*.

these studies, we first reviewed the proteins and identified 46 proteins that were detected in all the studies (*Figure 4A*). We then examined their abundances in the 205 cancer and 51 control samples (*Figure 4B*). Although we could detect differences between the exosomes derived from cancer and non-cancer samples, it was difficult to classify them robustly based on PCA (*Figure 4—figure supplement 1A*). Therefore, we sought to employ the advanced machine learning algorithm to differentiate cancer exosomes from non-cancer exosomes. We first calculated the mutual information (MI) score for each protein and trained the random forest classifier using the different numbers of top proteins according to their MI score to determine the best set of proteins we should include in the classifier. We found that the model performed best using 18 proteins, where the performance was evaluated by the area under the curve of the receiver operating characteristic curve (AUROC; *Figure 4C*). Several key cancer-associated proteins were included among these 18 proteins. For example, apolipoprotein C1 (APOC1), which was ranked as the top protein, is significantly decreased in samples of cancer patients (*Figure 4—figure supplement 1B*) and has been previously reported to be down-regulated in the non-small lung cancer, colorectal cancer, papillary thyroid carcinoma, and pediatric nephroblastoma (*Zhang et al., 2011*; *Jin et al., 2019*; *Engwegen et al., 2008*; *Fan et al., 2009*).

Employing all 18 proteins with five-fold cross-validation, we constructed a random forest classifier (*Ho, 1995*) to distinguish cancer and control samples and compared it with multiple popular machine learning models, including Support Vector Machine (SVM; *Cortes and Vapnik, 1995*), K Nearest Neighbor Classifier (K-NN; *Fix and Hodges, 1989*), and Gaussian Naive Bayes (*Pedregosa, 2011*). Our random forest classifier demonstrated the highest AUROC (*Figure 4D*). More specifically, it yielded an AUROC of 0.96, with an accuracy of 0.92, a precision of 0.94, and a recall of 0.96 (*Figure 4E and F*). When applied to the independent test set, the model yielded an AUROC of 0.99, an accuracy of 0.95, a precision of 0.96 and a recall of 0.98 (*Figure 4G and H*). Importantly, only one sample was misclassified in the independent test set, and 51 cancer samples were correctly classified. Taken together, our results showed the advantage and clinical potential of applying the random forest classifier model to plasma or serum exosome protein based liquid biopsy for cancer diagnosis.

## Five plasma/serum exosomal proteins can reliably differentiate five common cancer types

We next sought to further enhance the clinical utility of the exosomes for differentiating cancer types. We analyzed proteomics data from plasma or serum-derived exosomes from patients across five common cancer types, including breast cancer, colorectal cancer, glioma, lung cancer, and pancreatic cancer. Initial PCA revealed differences in exosome levels among cancer patients but failed to distinguish the five cancer types (*Figure 5A*). We determined the crucial features for cancer-type classification by computing mutual information scores for 46 common proteins and built a random forest classifier to determine the optimal number of features to include in the final classifier. We ultimately selected a set of five proteins based on AUROC scores (*Figure 5B*). We then increased the independent testing data size by utilizing 40% of the total samples and used the remaining 60% as the training set to minimize overfitting issues. Employing the five proteins with five-fold cross-validation

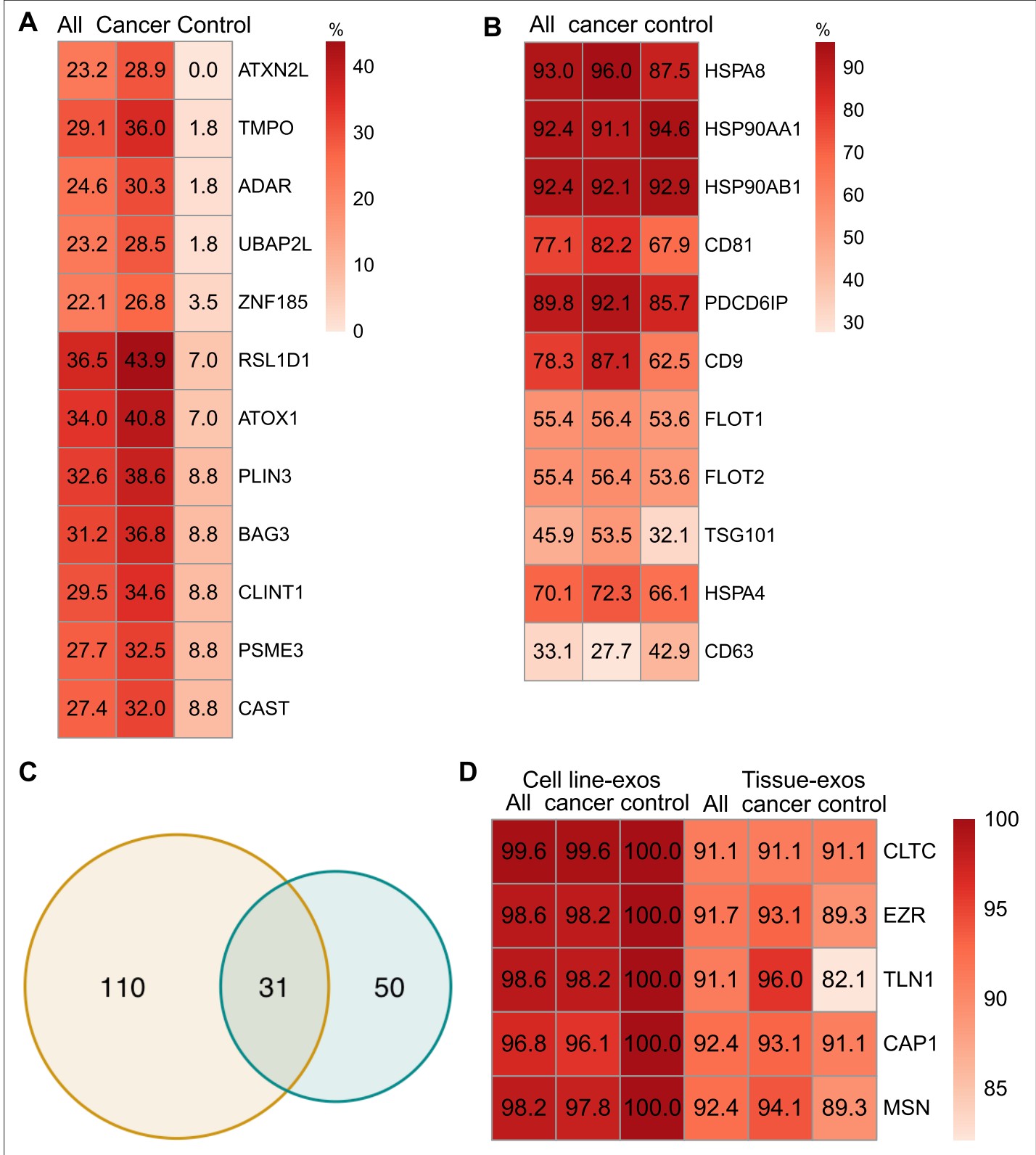

**Figure 3.** Proteomic characterization of exosomes derived from cell lines and tissues. (**A**) Proteins detected at higher frequency in cancer cell line-derived exosomes. (**B**) Positivity for 11 commonly used exosomal protein biomarkers in various tissues. (**C**) Overlapping proteins (>90% frequency) between cell line- and tissue-derived exosomes. (**D**) Positivity of five plasma membrane proteins detected in more than 90% of both cell line- and tissue-derived exosomes.

*Figure 3 continued on next page*

*Figure 3 continued*

The online version of this article includes the following source data for figure 3:

**Source data 1.** Related to *Figure 3A, B and D*.

**Source data 2.** Related to *Figure 3C*.

to train the random forest classifier, the model achieved a very high accuracy of 0.99 (*Figure 5C*), and when applied to the independent test set, the model consistently yielded a high accuracy of 0.94 (*Figure 5D*). The abundance of the five protein features varied across the five cancer types, reflecting the potential roles of these proteins in specific cancers (*Figure 5E*). For example, Histidine-rich glyco-protein (HRG), which was abundant in the colorectal cancer plasma-derived exosomes, has been reported to promote the tumor migration of colorectal cancer patients (*Bogoevska et al., 2017*). Overall, our results demonstrated that this exosome protein-based classification model can reliably differentiate between cancer types and further enhance the diagnostic value of our approach.

## A urinary exosome proteome signature consisting of 17 proteins detects cancer exosomes across multiple cancer types

Urine is emerging as a superior non-invasive marker for urologic cancers, as its composition directly reflects the physiological changes in the urogenital system (*Dhondt et al., 2020*). To test the use of urinary exosome proteins in cancer diagnosis, we collected data from 261 cancer patient samples and 124 control samples from four studies, including bladder cancer, prostate cancer, and renal cancer, lung cancer, cervical cancer, colorectal cancer, esophageal and gastric cancer (*Supplementary file 1*; *Dhondt et al., 2020*; *Zhang et al., 2018*; *Øverbye et al., 2015*; *Suh et al., 2022*). Upon examination of proteins common across all four studies, 229 proteins were identified (*Figure 6A*). PCA revealed the variance between samples but failed to differentiate between cancer and control samples (*Figure 6B*). As described earlier, we next employed the random forest classifier to differentiate cancer and control samples based on their exosomal proteomic profiles. To reduce the feature space and select the most relevant features, we utilized the mutual information score to rank the 229 protein features and then trained the random forest classifier using varying numbers of the top-ranking proteins (*Figure 6C*). Based on the AUROC scores of including the different number of features, we selected 17 features that resulted in the highest AUROC score (*Figure 6C*). A majority of these 17 proteins displayed significant variations in abundance between cancer and control samples (*Figure 6D*). By training a random forest classifier with the 17 protein features and five-fold cross-validation, the model achieved an AUROC of 0.96, an accuracy of 0.90, a precision of 0.92, and a recall of 0.93 (*Figure 6E and F*). When tested on an independent set, the model produced an AUROC of 0.91, an accuracy of 0.82, a precision of 0.83, and a recall of 0.92 (*Figure 6G and H*). To summarize, our findings indicated the promising clinical potential of using urinary exosome proteins for the diagnosis of urologic cancers as well as other non-urologic cancer types.

## Discussion

Liquid biopsy has numerous benefits in the early detection of cancer, categorizing cancer types, tracking cancer progression, and monitoring response to treatment (*Nikanjam et al., 2022*). Exosomes found in biological fluids can provide a forensic view of their cells of origin. Despite previous studies proposing common protein biomarkers to identify exosomes, a comprehensive set of exosome biomarkers derived from different biological materials has not been established, owing to limita-tions in isolation and quantification methods (*Castillo et al., 2018*; *Gangoda et al., 2017*; *Hurwitz et al., 2016*; *Ji et al., 2013*). Additionally, a reliable diagnostic tool based on proteins associated with exosomes that can be applied across all cancers is yet to be identified.

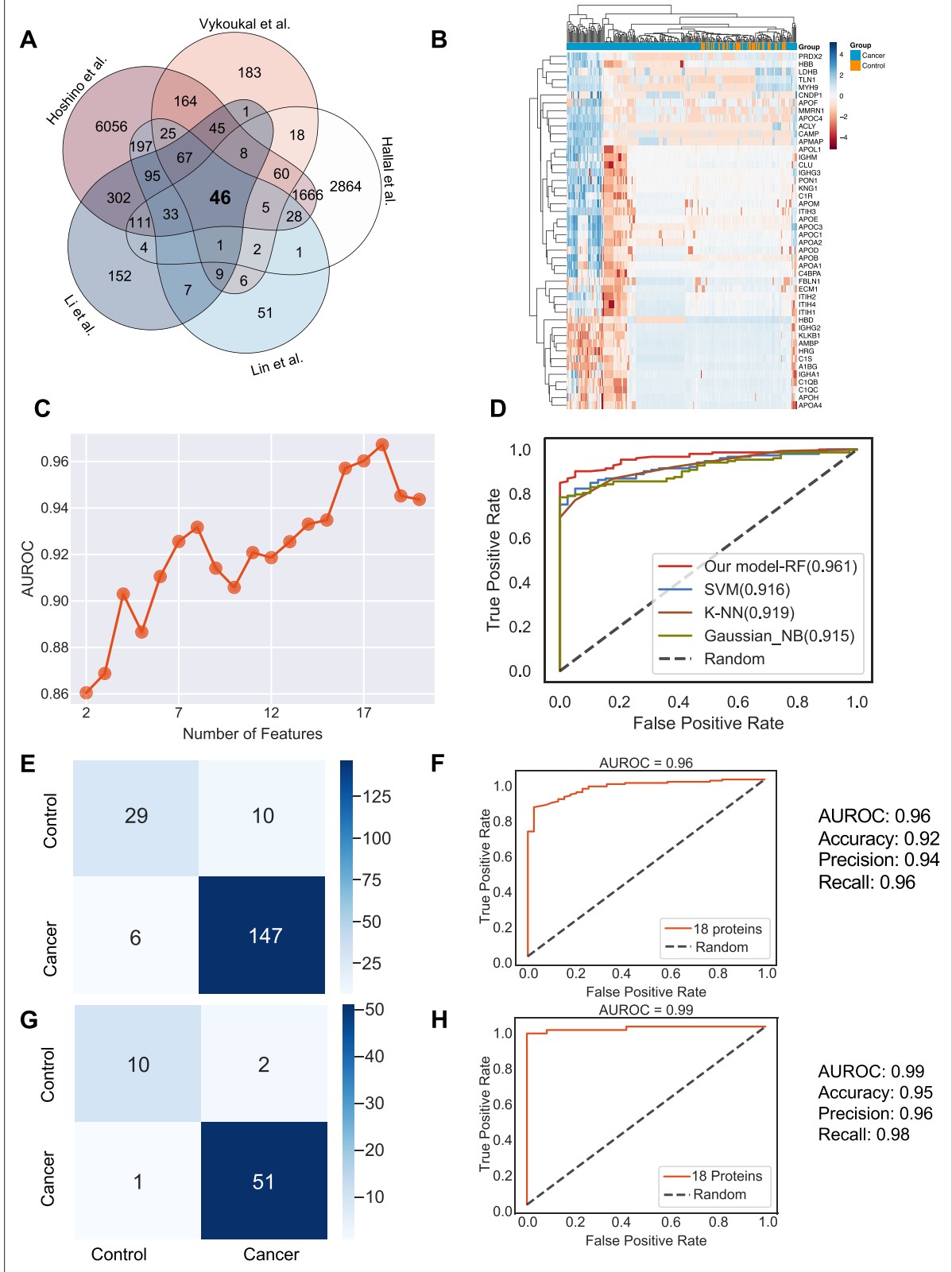

**Figure 4.** Identification of the signature proteins of plasma or serum-derived exosomes and the evaluation of random forest classifier. (**A**) Overlapping exosome proteins detected in the plasma and serum of 205 cancer and 51 control samples from five studies. (**B**) Heat map of 46 overlapping exosome proteins in cancer and control plasma or serum samples. (**C**) AUROC score of the random forest classifier on including various numbers of protein features. (**D**) AUROC of different models in comparison. (**E**) Classification error matrix of the 75% training set using a random forest classifier for the 18

*Figure 4 continued on next page*

*Figure 4 continued*

selected proteins. The number of samples is indicated in each box. (**F**) AUROC score of the random forest classifier trained using 75% of the dataset. Other metrics are indicated on right. (**G**) Classification error matrix of 25% testing set using a random forest classifier for the 18 selected proteins. The number of samples is indicated in each box. (**H**) AUROC score of the random forest classifier tested using 25% of the dataset. Other metrics are indicated on right.

The online version of this article includes the following source data and figure supplement(s) for figure 4:

**Source data 1.** Related to *Figure 4*.

**Figure supplement 1.** Machine learning models for plasma-derived exosomes.

Here, we generate a comprehensive proteomics profile of exosomes derived from cell lines, tissues, plasma, serum, and urine from 1083 cancer and control samples. An extensive analysis of these samples showed that several widely used exosome markers, such as CD63, CD81, HSP70, and HSPA8, are absent in exosomes derived from a subset of cell lines. Further, FLOT1, FLOT2, TSG101, and CD63 are present at low levels (<60%) in all exosomes derived from tissues, indicating a need for the identification of additional universal markers for both cell line- and tissue-derived exosomes. Our study identifies five highly abundant universal exosome biomarkers-CLTC, EZR, TLN1, CAP1, and MSN- that were present in over 90% of all cell line- and tissue-derived samples. Additionally, we found that ATXN2L was only present in cancer exosomes and absent in non-cancer exosomes. In this regard, GPC1 which was identified in many studies identified as cancer exosomes specific marker, was not identified here as many of datasets did not pick up this protein in analysis.

Here, we describe a novel computational approach using the random forest classifier method to define exosome protein panels that serve as effective biomarkers specifically for plasma, serum, or urine across cancer types. By training the random forest model and testing with independent data-sets, our model yields excellent scores in AUROC, sensitivity, and specificity for differentiating cancer exosomes from non-cancer exosomes. We show that this approach can also be used to classify, with high accuracy, five common cancer types based on their exosome protein signatures.

A majority of the protein makers identified in this study have demonstrable biological relevance in cancer. As an example, ITIH3, which was identified among the protein features for the plasma or serum-based classifier and highly abundant in cancer samples, was reported to be more highly expressed in the plasma of gastric cancer samples compared to the control (*Chong et al., 2010*) and increased with tumor staging of clear cell renal cell carcinoma patients (*Chang et al., 2021*). Importantly, the biomarker panels identified for cell lines, tissues, plasma/serum, and urine overcome the bias associated with the isolation and quantification method as well as the inter-patient variability inherent to the complex process of cancer. These panels are designed to be applied to a variety of cancers.

Collectively, our results demonstrate that exosome protein features can be utilized as reliable biomarkers for the early detection of cancer, classification of cancer types, and potentially for diag-nosing tumors of undetermined origin. These results have the potential to advance the develop-ment and standardization of innovative and optimized methods for the isolation of exosomes and the implementation of routine plasma, serum- and urine-based exosome screening in clinical settings.

## Methods
### Public exosome proteomics data

We collected publicly available exosome protein data from cell lines, plasma, serum, and urine from previous studies as summarized in *Hoshino et al., 2020*; *Vykoukal et al., 2017*; *Li et al., 2021*; *Lin et al., 2022*; *Hallal et al., 2020*; *Dhondt et al., 2020*; *Zhang et al., 2018*; *Øverbye et al., 2015*; *Suh et al., 2022*; *Hurwitz et al., 2016*; *Rontogianni et al., 2019*; *Supplementary file 1*. We obtained

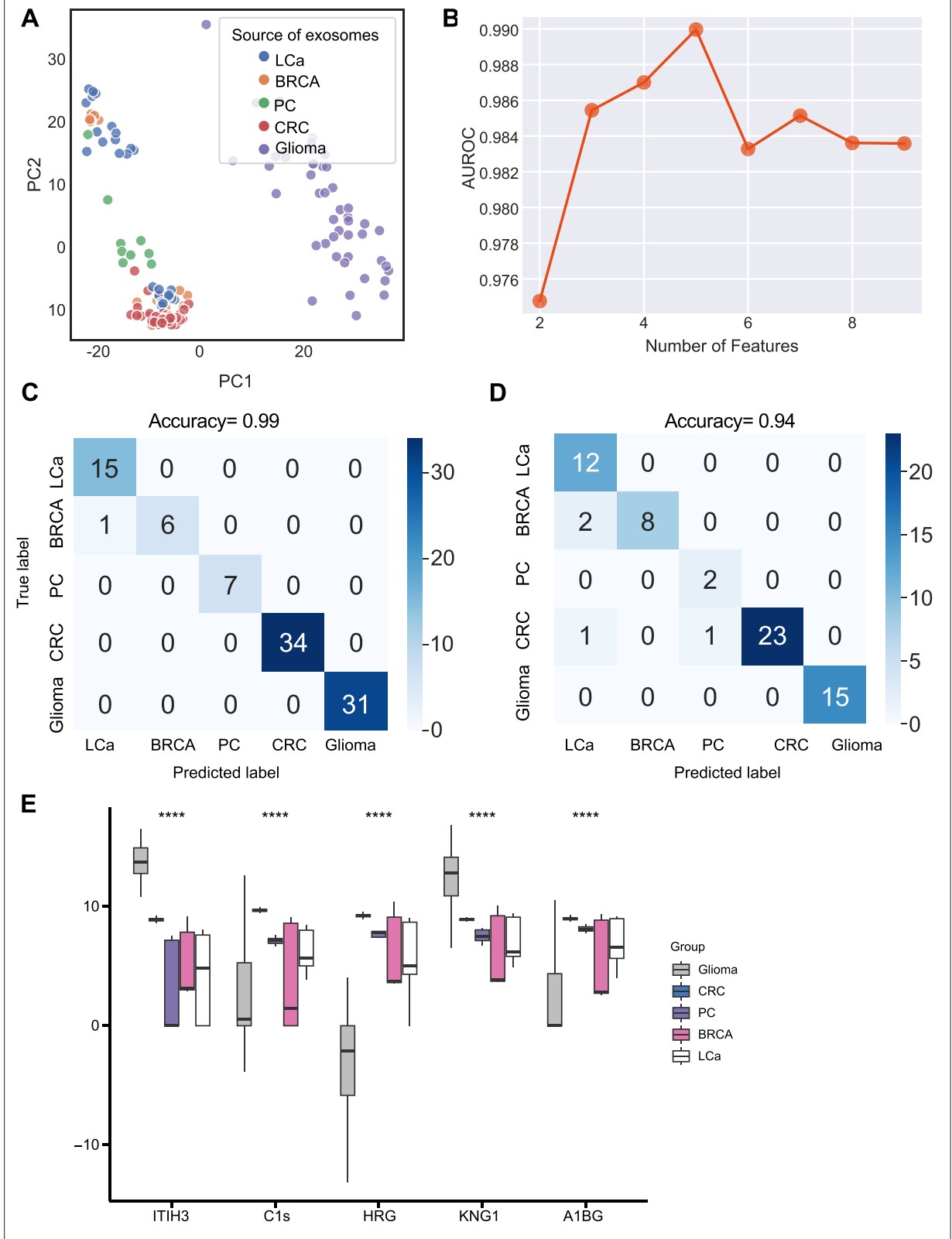

**Figure 5.** Identification of signature proteins expressed by plasma or serum-derived exosomes for classifying five common cancer types and evaluation of random forest classifier. (**A**) PCA plot of plasma or serum-derived exosomes from five cancer types. (**B**) AUROC score of the random forest classifier by including various number of protein features. (**C–D**) Classification error matrix of a 60% training set and 40% testing set to classify the five cancer types

*Figure 5 continued on next page*

*Figure 5 continued*

using a random forest classifier for the five selected proteins. The number of samples is indicated in each box. (**E**) Protein abundance of five selected protein features in 158 samples across five cancer types. Significance was determined by the Kruskal-Wallis test. ****p < 0.0001.

The online version of this article includes the following source data for figure 5:

**Source data 1.** Related to *Figure 5*.

the raw spectral count and intensity data from the original paper or directly from the authors through communication. The obtained data was log normalized accordingly.

## Feature selection and machine learning algorithm

We employed mutual information scores to evaluate the importance of protein features for each prediction, quantifying the amount of uncertainty reduction for one variable given the knowledge of another variable. We calculated the mutual information score for each protein with a target label (cancer/control) using SelectKBest from the scikit-learn 1.1.1 Python library.

The mutual information score quantifies the decrease in uncertainty of one variable when the value of the other variable is known. We performed the mutual information score calculation twenty times and computed the average score for each protein as the ultimate score. According to the sample size and characteristics of the data collected for plasma and urine-derived exosomes, we selected a customized number of best features for each prediction. To select the optimum number of proteins to include in the prediction, we ranked the protein features based on the mutual information score and built a random forest classifier to evaluate performance on including a range of number of features.

The random forest model can decrease the probability of over-fitting and enhance the resilience towards outliers and input data noise. The area under the curve of the receiver operating characteristic curve (AUROC) was employed to evaluate the performance of the classifier. We also calculated accuracy, precision and recall for comprehensive evaluation. All models were evaluated using five-fold cross-validation with stratified train-test splits that preserved the percentage of samples for the prediction target. We also tested the performance of alternate machine learning algorithms including support vector classifier, K nearest neighbor classifier and Gaussian Naive Bayes. Overall, the random forest classifier achieved the best performance in our analysis (*Figure 4D*). To visualize high-dimensional datasets, kernel PCA from scikit-learn 1.1.1 Python library was employed to perform PCA and plots were generated using the scatterplot function from seaborn 0.11.2 Python library.

## Gene ontology and pathway enrichment analysis

The WebGestalt 2019m (*Liao et al., 2019*) online tool was used to perform the gene ontology and pathway enrichment analysis of the selected proteins. The biological process, cellular component and molecular function were all selected for gene ontology analysis and the Kyoto Encyclopedia of Genes and Genomes Pathways database was selected for pathway analysis. with FDR <0.05 were considered significant. IPA analysis was conducted by QIAGEN Ingenuity Pathway Analysis software. The protein list was uploaded to the software and then annotated with the default settings.

## Statistical analysis

All statistical analyses were conducted using R 4.2.1 software. Significance was determined by the Wilcoxon rank-sum test unless specified otherwise. Significance was concluded if the p-value was <0.05, while in pathway analysis, significance was concluded if the FDR was <0.05 after correction for multiple comparisons.

## Acknowledgements

The EV work in the Kalluri lab is supported by MD Anderson Cancer Center, NIH R35CA263815, and NIH P40OD024628 and gifts from Fifth Generation (Love, Tito's), Lyda Hill Philanthropies, and Bosarge Family Foundation.

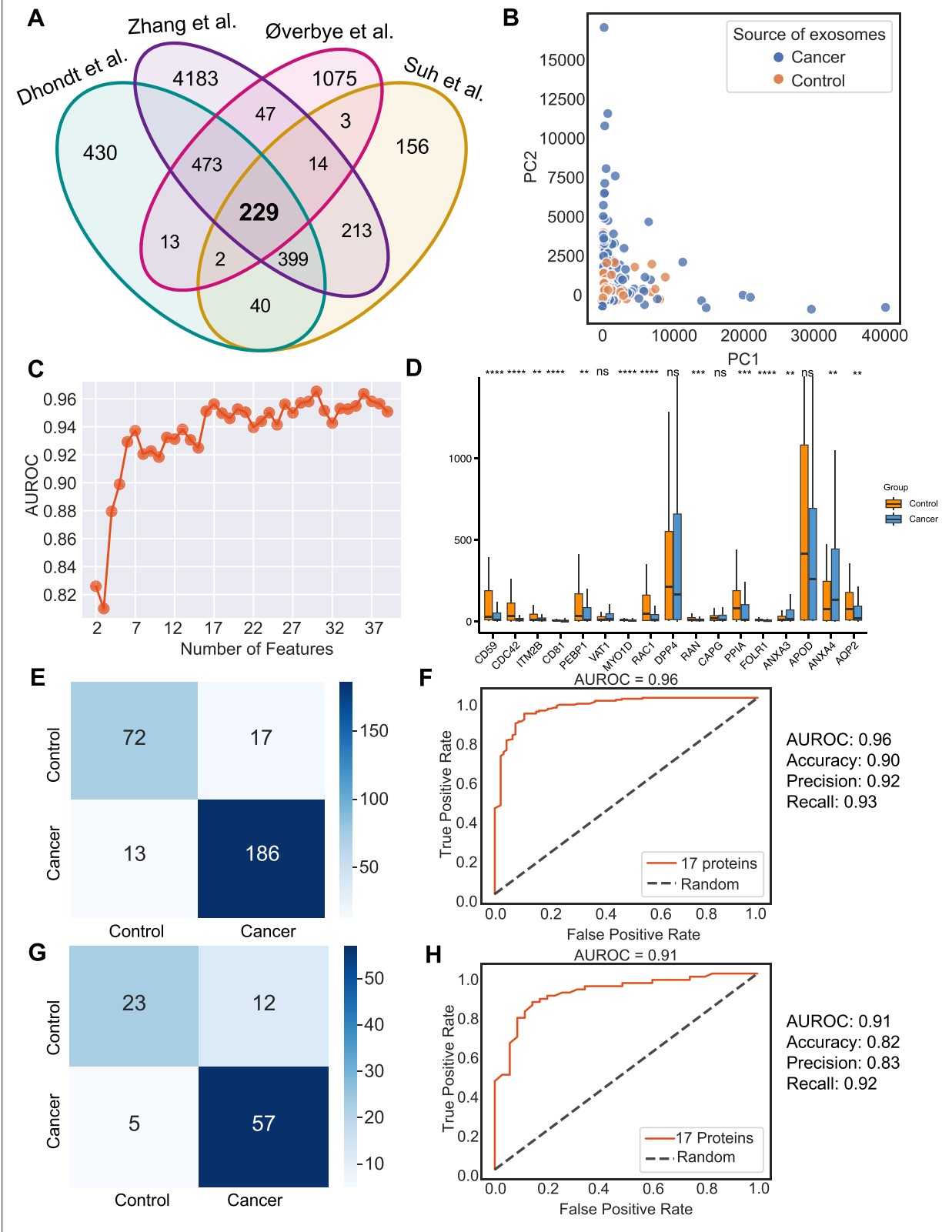

**Figure 6.** Identification of signature proteins expressed by urine-derived exosomes and evaluation of random forest classifier. (**A**) Overlapping exosome proteins detected in the urine from 261 cancer and 124 control samples from four studies. (**B**) PCA plot of cancer and control urine-derived exosomes. (**C**) AUROC score of the random forest classifier by including a various number of protein features. (**D**) Protein abundance of 17 selected protein features in 261 cancer- and 124 control urine-derived exosomes. (**E**) Classification error matrix of 75% training set using a random forest classifier for the 17

*Figure 6 continued on next page*

*Figure 6 continued*

selected proteins. The number of samples is indicated in each box. (**F**) AUROC score of the random forest classifier trained using 75% of the dataset. Other metrics are indicated on right. (**G**) Classification error matrix of 25% testing set using a random forest classifier for the 17 selected proteins. The number of samples is indicated in each box. (**H**) AUROC score of the random forest classifier tested using 25% of the dataset. Other metrics are indicated on the right. Significance was determined by the Wilcoxon rank-sum test. **p < 0.01, ***p < 0.001, ****p < 0.0001, *ns*: not significant.

The online version of this article includes the following source data for figure 6:

**Source data 1.** Related to *Figure 6*.

# Additional information

### Competing interests

Raghu Kalluri: UT MD Anderson Cancer Center and RK hold patents in the area of exosome biology licensed to Codiak Biosciences, Inc; UT MD Anderson Cancer Center and RK are stock equity holders in Codiak Biosciences, Inc; RK is a consultant and scientific adviser for Codiak Biosciences, Inc. The other authors declare that no competing interests exist.

### Funding

| Funder | Grant reference number | Author |
|---|---|---|
| University of Texas MD Anderson Cancer Center | | Raghu Kalluri |
| National Institutes of Health | R35CA263815 | Raghu Kalluri |
| National Institutes of Health | P40OD024628 | Raghu Kalluri |
| Fifth Generation (Love, Tito's) | | Raghu Kalluri |
| Lyda Hill Philanthropies | | Raghu Kalluri |
| Bosarge Family Foundation | | Raghu Kalluri |

The funders had no role in study design, data collection and interpretation, or the decision to submit the work for publication.

### Author contributions

Bingrui Li, Conceptualization, Data curation, Software, Formal analysis, Investigation, Visualization, Methodology, Writing – original draft, Writing – review and editing; Fernanda G Kugeratski, Data curation; Raghu Kalluri, Conceptualization, Supervision, Writing – original draft, Writing – review and editing

### Author ORCIDs

Bingrui Li ⓘ http://orcid.org/0000-0002-4807-1452
Raghu Kalluri ⓘ https://orcid.org/0000-0002-2190-547X

Reviewer #1 (Public Review): https://doi.org/10.7554/eLife.90390.3.sa1
Reviewer #2 (Public Review): https://doi.org/10.7554/eLife.90390.3.sa2
Author Response https://doi.org/10.7554/eLife.90390.3.sa3

# Additional files

### Supplementary files

- Supplementary file 1. Detailed information on the public data included in this study.
- Supplementary file 2. Proteins detected in more than in 90% of all cell line-derived exosomes.
- MDAR checklist
- Source code 1. The code of the random forest model developed in the study.

## Data availability

The current manuscript is a computational study, so no data have been generated for this manuscript. Modeling code is uploaded as *Source code 1*.

The following previously published datasets were used:

| Author(s) | Year | Dataset title | Dataset URL | Database and Identifier |
|---|---|---|---|---|
| Ji J, Li S | 2021 | Proteomic landscape of serum exosomes to identify potential therapeutic targets | https://proteomecentral.proteomexchange.org/cgi/GetDataset?ID=PXD027696 | ProteomeXchange, PXD027696 |
| de Wever O, Rappu P | 2020 | Unravelling the proteomic landscape of extracellular vesicles in prostate cancer by density-based fractionation of urine | https://proteomecentral.proteomexchange.org/cgi/GetDataset?ID=PXD015289 | ProteomeXchange, PXD015289 |
| Øverbye A, Skotland T, Koehler CJ, Thiede B, Seierstad T, Berge V, Sandvig K, Llorente A | 2015 | Identification of prostate cancer biomarkers in urinary exosomes | https://proteomecentral.proteomexchange.org/cgi/GetDataset?ID=PXD090912 | ProteomeXchange, PXD090912 |
| Molina H, Lyden DC | 2020 | Tissue- and plasma-derived exosomal protein biomarkers define multiple human cancers | https://www.ebi.ac.uk/pride/archive/projects/PXD018301 | PRIDE, PXD018301 |

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
