## [Editor Report · eLife assessment]

This **important** study introduces a novel AI method for the analysis of published data, with practical implications for early cancer diagnosis. The results are supported by **compelling** evidence.

---

## [Referee Report · Reviewer #1 (Public Review)]

Summary:

In this manuscript, the authors used machine learning algorithm to analyze published exosome datasets to find biomarkers to differentiate exosomes of different origin. By applying the method to "exosomes" sample, the author discovered common exosome markers and cancer-type specific markers.

Strengths:

The performance of the algorithm are generally of good quality.

---

## [Referee Report · Reviewer #2 (Public Review)]

Summary:

This is a fine work on the development of computational approaches to detect cancer through exosomes. Exosomes are an emerging biomarker resource and have attracted considerable interests in the biomedical field. Kalluri and co-workers collected a large sample pool and used random forest to identify a group of protein markers that are universal to exosomes and to cancer exosomes. The results are very exciting and not only added new knowledge in cancer research but also a new and advanced method to detect cancer. Data was presented very nicely and the manuscript was well written.

Strengths:

Identified new biomarkers for cancer diagnosis via exosomes.

Developed a new method to detect cancer noninvasively.

Results were presented nicely and manuscript were well written.

---

## [Author Response]

The following is the authors’ response to the original reviews.

**Reviewer #1:**
Summary:In this manuscript, the authors used machine learning algorithm to analyze published exosome datasets to find biomarkers to differentiate exosomes of different origin.Strengths:The performance of the algorithm are generally of good quality.Weaknesses:The source datasets are heterogeneous as described in Figure 1 and Figure 2, or Line 72-75; and therefore questionable.

Response: We thank the reviewer for this assessment. The commonly used biomarkers of exosomes exhibit heterogeneous presence and abundance within the exosomes derived from different cell lines, tissue, and biological fluids. The primary goal of this study was to identify universal exosomal biomarkers that remain consistent across different sources of exosomes, unaffected by potential isolation and quantification bias. This objective was achieved through an integration of datasets from different sources, which allowed for the subsequent identification of common proteins associated with exosomes. Among the 18 protein markers identified, it is noteworthy that they are universally abundant in all cell lines and their exosomes. We believe that despite the heterogeneity of the datasets used here, the identification of 18 universal protein markers in exosomes from diverse sources is a strength of this analysis.

(1) Nomenclature: Extracellular vesicles (EVs) are small vesicles released by cells into the extracellular space, exhibiting high heterogeneity in origin across species. Exosomes are typically defined as being of multivesicular body origin. However, the absence of several crucial common exosomal markers, including CD63, suggests that the proteomics analysis may include various other vesicular and non-vesicular materials.

Response: As we reported previously (Kugeratski et al., Nature Cell Biology, 2021), the commonly used exosomal markers, such as CD9, CD63 and CD81 exhibit heterogeneity with respect to presence and abundance in the exosomes derived from different cell types. For example, CD63 demonstrated remarkably lower abundance in the exosomes derived from Raji cell lines. In our study, the detection rate of CD63 (< 50%) is quite low in the tissue-derived exosomes, which is consistent with the observations made in another proteomics based study (Hoshino et al., Cell, 2020). Therefore, relying solely on these markers is inadequate for the comprehensive characterization of EVs as exosomes. Therefore, we conducted this study to identify universal protein markers of exosomes by integrating data from multiple sources, thereby circumventing potential confounding effects due to their diverse origins and other technical differences.

(2) Line 90: IPA is not prior in the manuscript.

Response: We provided the full definition of IPA (Ingenuity Pathway Analysis) in the revised manuscript.

(3) Figure 2B: Considering the large number of variables, it is unsurprising that the 2D PCA (Principal Component Analysis) falls short in the classification task. Including a few additional dimensions (principal components) might have the potential to better distinguish the cancer groups from the control group.

Response: Thank you for this insightful query. The purpose of utilizing PCA here is to appreciate the heterogeneity associated with exosomes from different studies. While we acknowledge that additional dimensions may be more useful in distinguishing between cancer and control exosomes, we believe that derived performance will remain inferior to the machine learning approach we developed here.

(4) Figure 2D: Exosomes primarily derive from multivesicular bodies, rather than the plasma membrane. It remains unclear why the authors focus specifically on proteins in the plasma membrane. Is it intended to encompass all membrane proteins? Clarification is needed on this point.

Response: A good point. This study attempted to identify protein biomarkers of exosomes originating from different sources. Our approach involved considering proteins present on the plasma membrane as potential biomarkers also because many of them have been detected on the surface of exosomes.

(5) Figure 2F: The 18 identified proteins are also abundantly present in control cells, not solely in cancer-derived "exosomes." The statement in line 104 is misleading in this regard.

Response: We apologize for the misleading sentence. We have revised the statement to state that “In total, we identified a set of 18 exosome protein markers that are present at a higher abundance in all exosomes examined”.

(6) Figure 3B: Considering the definition of exosomes, CD63 and TSG101 should be present in all samples, and their absence raises concerns.

Response: We understand the concern of the reviewer. In this Figure, we analyzed CD63 and TSG101 in tissue-derived exosomes. Our results are consistent with the previous study also shows the paucity of these makers in the tissue-derived exosomes (Hoshino et al., Cell, 2020). Our study highlights that CD63 and TSG101 cannot always identify exosomes from diverse cell lines and tissues. Such initial observations motivated us to conduct this study to identify the universal biomarkers of exosomes across different sources.

(7) Figure 6G&H: Achieving an accuracy of 80% cannot be deemed "excellent."

Response: We employed the word “excellent” in line 225 to describe the sensitivity and specificity associated with AUROC.

(8) Other comments on methods: The manuscript lacks an explanation of the neural network structure and why it outperforms other methods. Additionally, details about the calculation of MI (mutual information), IPA, and other methods should be provided.

Response: This is a good suggestion but in this work we did not employ the neural networks for the analysis. We provided additional details and explanations regarding the methodology for mutual information score calculation, as well as insights into the improved use of IPA and other relevant methods in the revised manuscript.

**Reviewer #2:**
Summary:This is a fine work on the development of computational approaches to detect cancer through exosomes. Exosomes are an emerging biomarker resource and have attracted considerable interests in the biomedical field. Kalluri and co-workers collected a large sample pool and used random forest to identify a group of protein markers that are universal to exosomes and to cancer exosomes. The results are very exciting and not only added new knowledge in cancer research but also a new and advanced method to detect cancer. Data was presented very nicely and the manuscript was well written.Strengths:Identified new biomarkers for cancer diagnosis via exosomes.Developed a new method to detect cancer non-invasively.Results were presented nicely and manuscript were well written.Weaknesses:N/A.

Response: We appreciate the the enthusiastic assessment of our study by the reviewer.

**Reviewer #3:**
In the current study, Li et al. address the difficulty in early non-invasive cancer diagnosis due to the limitations of current diagnostic methods in terms of sensitivity and specificity. The study brings attention to exosomes - membrane-bound nanovesicles secreted by cells, containing DNA, RNA, and proteins reflective of their originating cells. Given the prevalence of exosomes in various biological fluids, they offer potential as reliable biomarkers. Notably, the manuscript introduces a new computational approach, rooted in machine learning, to differentiate cancers by analyzing a set of proteins associated with exosomes. Utilizing exosome protein datasets from diverse sources, including cell lines, tissues, and various biological fluids, the study spotlights five proteins as predominant universal exosome biomarkers. Furthermore, it delineates three distinct panels of proteins that can discern cancer exosomes from non-cancerous ones and assist in cancer subtype classification using random forest models. Impressively, the models based on proteins from plasma, serum, or urine exosomes achieve AUROC scores above 0.91, outperforming other algorithms such as Support Vector Machine, K Nearest Neighbor Classifier, and Gaussian Naive Bayes. Overall, the study presents a promising protein biomarker signature tied to cancer exosomes and proposes a machine learning-driven diagnostic method that could potentially revolutionize non-invasive cancer diagnosis.

Response: We appreciate this positive assessment of our work.

(1) The authors should clarify why they focused solely on protein markers. Why weren't RNA transcripts also considered? Do the authors see value in incorporating RNA/micro RNA transcripts to enhance diagnostic capabilities?"

Response: This is a very important point for further consideration. The current datasets for exosomal proteins are extensive and generally proteins might offer distinct advantages in cancer diagnostics compared to nucleic acids due to their stability in exosomes and extended half-life (Schey et al., Methods, 2015). We do agree that the power of analysis can only get better if also add DNA, RNAs and other constituents and we hope to pursue such analysis in the future.

(2) Can the identified exosomal markers also be evaluated as prognostic indicators?

Response: We appreciate this intriguing question. Indeed, proteins such as apolipoprotein E (APOE) may serve as a potential prognostic marker in various cancers (Ren et al., Cancer Medicine, 2019). APOE is being extensively studied as a prognostic and diagnostic marker for multiple cancer types, including colorectal cancer (Martin et al., BMC Cancer, 2014), gastric cancer (Sakashita et al., Oncology Reports, 2008), pancreatic cancer (Chen et al., Medical Oncology, 2013; Xu et al., Tumor Biology, 2016), and human hepatocellular carcinoma (Yokoyama et al., International Journal of Oncology, 2006). In these studies, APOE levels were found to be elevated in the serum of cancer patients and correlated with survival outcomes.

(3) The discussion should emphasize if the identified protein markers are tumor-specific or if they indicate, for instance, the patient's immune reaction to the tumor.

Response: A good point. We believe that the identified biomarkers are tumor-specific and a significant number of these proteins have been previously associated with tumor initiation and progression. Further studies will likely identify immune response-related biomarkers when more in-depth tumor-level analyses are performed.

References:

Chen, J., Chen, L. J., Yang, R. B., Xia, Y. L., Zhou, H. C., Wu, W., Lu, Y., Hu, L. W., & Zhao, Y. (2013). Expression and clinical significance of apolipoprotein E in pancreatic ductal adenocarcinoma. Med Oncol, 30(2), 583. https://doi.org/10.1007/s12032-013-0583-y

Hoshino, A., Kim, H. S., Bojmar, L., Gyan, K. E., Cioffi, M., Hernandez, J., Zambirinis, C. P., Rodrigues, G., Molina, H., Heissel, S., Mark, M. T., Steiner, L., Benito-Martin, A., Lucotti, S., Di Giannatale, A., Offer, K., Nakajima, M., Williams, C., Nogues, L., . . . Lyden, D. (2020). Extracellular Vesicle and Particle Biomarkers Define Multiple Human Cancers. Cell, 182(4), 1044-1061 e1018. https://doi.org/10.1016/j.cell.2020.07.009

Kugeratski, F. G., Hodge, K., Lilla, S., McAndrews, K. M., Zhou, X., Hwang, R. F., Zanivan, S., & Kalluri, R. (2021). Quantitative proteomics identifies the core proteome of exosomes with syntenin-1 as the highest abundant protein and a putative universal biomarker. Nat Cell Biol, 23(6), 631-641. https://doi.org/10.1038/s41556-021-00693-y

Martin, P., Noonan, S., Mullen, M. P., Scaife, C., Tosetto, M., Nolan, B., Wynne, K., Hyland, J., Sheahan, K., Elia, G., O'Donoghue, D., Fennelly, D., & O'Sullivan, J. (2014). Predicting response to vascular endothelial growth factor inhibitor and chemotherapy in metastatic colorectal cancer. BMC Cancer, 14, 887. https://doi.org/10.1186/1471-2407-14-887

Ren, L., Yi, J., Li, W., Zheng, X., Liu, J., Wang, J., & Du, G. (2019). Apolipoproteins and cancer. Cancer Med, 8(16), 7032-7043. https://doi.org/10.1002/cam4.2587

Sakashita, K., Tanaka, F., Zhang, X., Mimori, K., Kamohara, Y., Inoue, H., Sawada, T., Hirakawa, K., & Mori, M. (2008). Clinical significance of ApoE expression in human gastric cancer. Oncol Rep, 20(6), 1313-1319. https://www.ncbi.nlm.nih.gov/pubmed/19020708

Schey, K. L., Luther, J. M., & Rose, K. L. (2015). Proteomics characterization of exosome cargo. Methods, 87, 75-82. https://doi.org/10.1016/j.ymeth.2015.03.018

Xu, X., Wan, J., Yuan, L., Ba, J., Feng, P., Long, W., Huang, H., Liu, P., Cai, Y., Liu, M., Luo, J., & Li, L. (2016). Serum levels of apolipoprotein E correlates with disease progression and poor prognosis in breast cancer. Tumour Biol. https://doi.org/10.1007/s13277-016-5453-8

Yokoyama, Y., Kuramitsu, Y., Takashima, M., Iizuka, N., Terai, S., Oka, M., Nakamura, K., Okita, K., & Sakaida, I. (2006). Protein level of apolipoprotein E increased in human hepatocellular carcinoma. Int J Oncol, 28(3), 625-631. https://www.ncbi.nlm.nih.gov/pubmed/16465366